# The Green Tea Polyphenol Epigallocatechin-Gallate (EGCG) Interferes with Microcin E492 Amyloid Formation

**DOI:** 10.3390/molecules28217262

**Published:** 2023-10-25

**Authors:** Paulina Aguilera, Camilo Berríos-Pastén, Marcelo Veloso, Matías Gálvez-Silva, Florian Turbant, Rosalba Lagos, Frank Wien, Veronique Arluison, Andrés E. Marcoleta

**Affiliations:** 1Grupo de Microbiología Integrativa, Laboratorio de Biología Estructural y Molecular BEM, Facultad de Ciencias, Universidad de Chile, Las Palmeras 3425 Ñuñoa, Santiago 7800003, Chile; paguilera@ug.uchile.cl (P.A.); camilo.berrios.p@ug.uchile.cl (C.B.-P.); marcelo.veloso@ug.uchile.cl (M.V.); matias.galvez.s@ug.uchile.cl (M.G.-S.); rolagos@uchile.cl (R.L.); 2Synchrotron SOLEIL, L’Orme des Merisiers, Saint Aubin BP48, 91192 Gif-sur-Yvette, France; florian.turbant@cea.fr (F.T.); frank.wien@synchrotron-soleil.fr (F.W.); 3Laboratoire Léon Brillouin LLB, CEA, CNRS UMR12, Université Paris-Saclay, CEA Saclay, 91191 Gif-sur-Yvette, France; 4UFR Sciences du Vivant, Université Paris Cité, 75006 Paris, France

**Keywords:** microcin E492, bacterial amyloid, EGCG, green tea, synchrotron-radiation circular dichroism

## Abstract

Microcin E492 (MccE492) is an antimicrobial peptide and proposed virulence factor produced by some *Klebsiella pneumoniae* strains, which, under certain conditions, form amyloid fibers, leading to the loss of its antibacterial activity. Although this protein has been characterized as a model functional amyloid, the secondary structure transitions behind its formation, and the possible effect of molecules that inhibit this process, have not been investigated. In this study, we examined the ability of the green tea flavonoid epigallocatechin gallate (EGCG) to interfere with MccE492 amyloid formation. Aggregation kinetics followed by thioflavin T binding were used to monitor amyloid formation in the presence or absence of EGCG. Additionally, synchrotron radiation circular dichroism (SRCD) and transmission electron microscopy (TEM) were used to study the secondary structure, thermal stability, and morphology of microcin E492 fibers. Our results showed that EGCG significantly inhibited the formation of the MccE492 amyloid, resulting in mainly amorphous aggregates and small oligomers. However, these aggregates retained part of the β-sheet SRCD signal and a high resistance to heat denaturation, suggesting that the aggregation process is sequestered or deviated at some stage but not completely prevented. Thus, EGCG is an interesting inhibitor of the amyloid formation of MccE492 and other bacterial amyloids.

## 1. Introduction

Amyloids are protein aggregates with a fibrillar morphology that are characterized by a hallmark cross-β structure consisting of β-sheets aligned perpendicularly to the fiber axis, creating a cross-like pattern [1]. These insoluble and remarkably stable structures share dye-binding and spectral properties that can be used for their detection, including a high affinity for the fluorescent probe Thioflavin T (ThT) and a circular dichroism spectral shift caused by increases in β-sheet content [2].

Traditionally, amyloids have been associated with neurodegenerative diseases such as Alzheimer’s (AD), Parkinson’s (PD), and Huntington’s [2], but over the last 20 years, several reports have shown amyloid assemblies playing biological roles. These “functional amyloids” appear to be present in all forms of life [3]. Moreover, they have been widely described in microorganisms and linked to several biological processes; functional amyloids may have roles in biofilm formation, niche colonization, and bacterial virulence, among other processes [4,5]. The first and best-understood example of a bacterial functional amyloid is curli, a type of extracellular amyloid fibers produced by *Escherichia coli* and *Salmonella* that act as an adhesive and structural support in biofilm formation [6]. Other examples related to biofilms include TasA produced by *Bacillus subtilis* [7], FapC produced by *Pseudomonas* spp. [8], and PSM produced by *Staphylococcus aureus* [9]. Besides biofilm formation, bacterial functional amyloids have been described to participate in aerial hyphae formation, cell cycle regulation, bacteria–plant symbiosis, and as acting as cytolytic toxins and antimicrobials [4,5].

Microcin E492 (MccE492) is a pore-forming bacteriocin produced by some *Klebsiella pneumoniae* strains which has antibacterial activity against related species [10]. The genetic determinants for its production are encoded in the GIE492 genomic island, which is highly prevalent among hypervirulent *K. pneumoniae* strains [11,12], although the specific role of MccE492 in virulence remains poorly understood. This bacteriocin is produced in two distinct forms: an unmodified form (7887 Da) and a post-translationally modified form, where glycosylated derivatives of the enterochelin siderophore are covalently linked to its C-terminus. This modification is required for antibacterial activity since the toxin internalization depends on its recognition by the catechol siderophore receptors of the target cells.

In addition to its antimicrobial activity, a remarkable feature of MccE492 is the ability to form amyloid aggregates in the cytoplasm and in the extracellular medium of producing cells [10,13,14,15]. This process has been studied for almost 20 years, making MccE492 a model bacterial amyloid [16]. In vitro, the aggregation of this peptide is typically observed in less than 24 h at 37 °C in 100 mM PIPES 0.5 M NaCl buffer (pH 6.5). In these conditions, the amyloid nature of the MccE492 fibers has been demonstrated by different approaches, including Congo Red and ThT binding, Fourier-transform infrared spectroscopy (FTIR), TEM, and X-ray diffraction [13,17,18]. The two latter approaches showed mainly ~140 Å-width helical fibers with a diffraction pattern having the two typical amyloid reflections at 9.9 and 4.7 Å, corresponding to the inter-strand and inter-sheet distances in the cross-beta array [14].

Interestingly, MccE492 amyloid formation is directly related to the loss of its antibacterial activity. This evidence has given rise to the hypothesis that MccE492 amyloid formation is a mechanism of antibacterial inactivation, forming a highly stable reservoir of antibacterial activity. Nevertheless, there is scarce information regarding the secondary structure transitions which occur as the MccE492 molecules adopt an amyloid-prone conformation. Moreover, there are no reports of molecules which possibly inhibit MccE492 amyloid formation. Given that MccE492 is secreted and can be recovered at a high purity from culture supernatants, it could be advantageous as a model protein to test for possible anti-amyloid molecules targeting bacterial functional amyloids.

Due to their relevance in neurodegenerative diseases and microbial virulence, substantial efforts have been made to identify molecules with anti-amyloid activity. Several natural compounds have been found to reduce the aggregation of Aβ40/42 associated to AD, especially polyphenols [19]. Epigallocatechin-3-gallate (EGCG) is a polyphenol found in green tea (*Camellia sinensis*) which has risen as an interesting anti-amyloid compound. Besides inhibiting Aβ40/42 amyloid formation and thus showing a potential therapeutic effect in AD, EGCG can interfere with α-synuclein aggregation and reduce its toxicity in PD [20,21]. Furthermore, EGCG also influences bacterial functional amyloids. In *S. aureus*, EGCG was shown to prevent the assembly of amyloidogenic PSM and the disaggregation of preformed amyloid fibers, converting them into amorphous aggregates [22]. Meanwhile, in *Pseudomonas* spp., EGCG inhibited FapC amyloid formation and stabilized non-amyloid aggregates [23]. Moreover, in *E. coli*, EGCG was shown to impair curli [24] and Hfq-CTR amyloid assembly and disrupt Hfq-CTR mature fibers [25]. In some of these cases, the inhibitory effect of EGCG on amyloid formation was related to bacterial virulence attenuation by inducing reduced bacterial survival [25] or impaired biofilm formation [23,24]. Thus, EGCG could also be explored as a potential anti-virulence agent for targeting amyloids involved in bacterial infection.

In this work, we assessed the effect of EGCG on MccE492 amyloid formation. Using a combination of techniques, including ThT fluorescence measurements, negative-staining transmission electron microscopy (TEM), and synchrotron radiation circular dichroism (SRCD), we investigated the secondary structure transitions and morphological signatures of MccE492 amyloid aggregation, providing orthogonal evidence that EGCG impairs MccE492 amyloid formation. This evidence supports that EGCG could be suitable for targeting amyloid-dependent bacterial processes in *K. pneumoniae*, particularly those related to MccE492.

## 2. Results

### 2.1. EGCG Inhibits MccE492 Amyloid Formation Followed by ThT Fluorescence

As a first approach to evaluating the effect of EGCG on MccE492 amyloid formation, we followed MccE492 aggregation kinetics in the absence and presence of EGCG using ThT, a widely used amyloid-binding fluorescent dye. For this purpose, we purified MccE492 from the culture supernatants of *E. coli* cells transformed with the pMccE492 plasmid which were carrying all the genes required for MccE492 synthesis, modification, and export [15]. Lyophilized purified MccE492 was reconstituted and mixed with the buffer that has been traditionally used to induce the aggregation of this peptide (100 mM PIPES pH = 6.5, 0.5 M NaCl) to a final protein concentration of 200 µg/mL. Besides the baseline reactions, two conditions were tested, one without EGCG (control) and the other with 1 mM EGCG. ThT was added to all conditions, and the assay was incubated with shaking at 37 ºC. The ThT fluorescence was measured every 15 s for 70 h.

Without EGCG, MccE492 showed typical sigmoidal amyloid aggregation kinetics, beginning with a lag phase where the formation of oligomers and the nucleation of small fibers occured (Figure 1A,B). After 12 h, the ThT fluorescence started to increase exponentially due to fiber elongation, reaching a plateau around 20 h once the assay started. Conversely, the ThT fluorescence remained close to the baseline levels in the presence of EGCG. A similar observation was made using a different buffer (100 mM sodium phosphate pH 6.5) that is more compatible with circular dichroism measurements and that could provide further evidence of the effect of EGCG, as described in Section 2.4. In this case, likely due to the decreased ionic strength, the lag phase was extended, and the aggregation started after 40 h, even when twice the MccE492 concentration (400 µg/mL) was used (Figure 1C). Again, with EGCG, no ThT fluorescence increase was observed. This evidence indicates that EGCG inhibited MccE492 amyloid formation.

### 2.2. MccE492 Is Not Incorporated into Higher-Order Aggregates in the Presence of EGCG

It has been suggested that EGCG and ThT binding to amyloid fibers compete with one another, and thus, the absence of ThT fluorescence in the presence of the polyphenol could be partly due to ECGC preventing the binding of ThT to the fibers. Thus, to obtain further evidence of the EGCG effect, we used a previously described assay to follow, over time, the MccE492 that remained unincorporated into fibers during aggregation, which was similar to that described in Section 2.1 but without the addition of ThT. At different time points, aliquots were collected and centrifuged (to pellet the fibers), and the supernatant was analyzed by immunoblot using a monoclonal anti-MccE492 antibody (Figure 2). Hence, as the aggregation progressed, less soluble microcin should have been observed.

Without EGCG, a similar amount of MccE492 remained in the solution during the first 12 h. At 24 h, the amount of protein in the supernatant markedly decreased, indicating that protein aggregation (i.e., amyloid formation) had occurred. Conversely, the protein remained soluble throughout the assay with 1 mM EGCG. Moreover, MccE492 tends to form SDS-resistant oligomers which are considered precursors to the amyloid fibers [14] (Figure 2). Interestingly, these oligomers are not observed in the presence of EGCG, suggesting that, besides inhibiting the formation of amyloid fibers, EGCG also inhibits the formation of amyloid precursors. These results confirm the inhibitory effect of EGCG on MccE492 amyloid formation as detected by ThT fluorescence.

### 2.3. EGCG Prevents the Formation of MccE492 Amyloid Fibers, as Revealed by Negative-Staining Electron Microscopy

Next, we aimed to directly visualize the effect of EGCG on MccE492 amyloid fiber formation. To this end, samples were collected at the beginning (0 h) and the end (48 h) of the aggregation assays, mounted, negatively stained, and visualized by Electron Transmission Microscopy (TEM) (Figure 3 and Appendix A). As expected, at 0 h, no fibers were observed independently of the presence of EGCG. As previously reported, only small amorphous aggregates were observed at this stage [14,15]. After 48 h, in the absence of EGCG, abundant amyloid fibers with the typical helical morphology [14,15] and an average width of 10 nm were detected. Conversely, in the presence of EGCG, no fibers were observed. This evidence confirms that EGCG completely inhibits the formation of MccE492 amyloid fibers.

### 2.4. EGCG Interferes with MccE492 Secondary Structure Transitions to Amyloid-Prone Forms

The MccE492 peptide was shown to have a partially disordered structure that shifts into a β-sheet-rich conformation during amyloid aggregation [18]. To investigate whether EGCG affects this secondary structure transition, we performed SRCD measurements in the DISCO vacuum UV beamline at the SOLEIL synchrotron (St. Aubin, France). As the PIPES aggregation buffer interferes with SRCD measurements, we prepared MccE492 aggregation assays in 100 mM phosphate buffer and increased the peptide concentration to 2 mg/mL, to shorten the lag phase and increase the SRCD signal, in a low-volume cell. MccE492 aggregated between 8 and 24 h after starting the assay in these conditions.

As expected, the SRCD spectra showed that, in the absence of EGCG, MccE492 progressively changed its conformation over time with a negative peak at ~218 nm, which is typical of the amyloid β-sheets found in the amyloid cross-beta structure (Figure 4A) [26]. In contrast, in the condition with EGCG, a markedly reduced signal around 218 nm was observed even 48 h after starting the assay (Figure 4B). This evidence indicates that EGCG impairs the transition of MccE492 into an amyloid-prone conformation.

We then used SRCD to evaluate the possible effect of EGCG on the secondary structure content and thermal stability of preformed MccE492 amyloid fibers. To this end, we performed thermal scans from 15 °C to 96 °C, collecting spectra every 3 °C (Figure 4C,D), followed by spectra deconvolution with BeStSel [27]. In this way, we could correlate the SRCD signal change due to the temperature increase with secondary structure changes. Previous reports indicate that MccE492 is highly resistant to temperature degradation, a property shared with other microcins [10,28]. Also, concerning EGCG, a previous study evaluated its thermal stability between 25 and 165 °C, showing that it remains mostly unaltered at 100 °C and a pH close to neutrality [29]. Thus, no considerable heat degradation of MccE492 nor EGCG was expected to occur in these assays.

At 15 °C, with or without EGCG, the aggregated MccE492 showed a parallel β-sheet content close to 40% (Appendix A). Following the temperature increase, without EGCG, the MccE492 amyloid showed a partial resistance to thermal denaturation, retaining part of the amyloid structure even at 96 °C (25.6% β-sheets). However, the β-sheet content was markedly lower in the presence of EGCG at 96 °C (9.5%). Moreover, in the absence of EGCG, a sustained decrease in beta-sheet content was observed at temperatures over 80 °C, while with EGCG, the loss of this structure was observed at lower temperatures (from 60 °C). Of note, an unexpectedly high alpha-helix content was observed at 15 °C, especially in the amyloid treated with EGCG (49.4%) and, to a lesser extent, the untreated peptide (30.6%). In this latter condition, upon heating, this structure fell to 1.2% at 96 °C, while it appeared more stabilized with EGCG (21.4% at 96 °C).

Thus, EGCG caused a partial reduction in the amyloid β-sheet signal and the α-helix content of preformed MccE492 amyloid, and decreased its thermal stability.

## 3. Discussion

Amyloid formation is a shared trait across important human neurodegenerative diseases. Considerable efforts have been made to discover anti-amyloid molecules that could work as a treatment. Bacterial functional amyloids may be an alternative simple model for developing and testing potential anti-amyloid molecules. Moreover, since some of these functional amyloids have also been related to bacterial virulence (i.e., biofilm formation, toxins reservoirs), they could also serve as a model to test anti-virulence agents [26]. MccE492 is one candidate protein that could serve this purpose, as it forms typical amyloid fibers in the extracellular space of the producing cells and, thus, can easily be purified from *E. coli* culture supernatants. Among the previously proposed anti-amyloid molecules, EGCG has been shown to prevent the amyloid formation of several human disease-associated and bacterial functional amyloid proteins. This work showed that EGCG also impairs MccE492 amyloid formation, with complementary evidence from ThT binding measurements, SDS-PAGE/immunoblot, TEM, and SRCD.

SRCD is a powerful spectroscopic technique that is suited for the detailed examination of amyloid proteins, as it enables precise characterization of their secondary structural components during the aggregation process and even their interactions with other macromolecules such as nucleic acids and lipid membranes [30,31,32,33,34]. By employing vacuum ultraviolet (VUV) light from synchrotron sources such as the DISCO beamline at SOLEIL [31], tunable and high-intensity circularly polarized light with exceptional sensitivity to molecular chirality and secondary protein structures can be obtained. Furthermore, low-noise spectra starting from 170 nm allow for well-fitted deconvolution.

Besides the EGCG effect, two noteworthy findings emerged from the SRCD and spectral deconvolution results. Firstly, a significant presence of α-helix secondary structure was observed, and secondly, the predicted β-sheets identified by the deconvolution model were of the parallel type. The bactericidal activity of MccE492 is associated with the formation of pores in the membranes of target bacteria, and it has been established that, within the membrane, MccE492 monomers adopt a transmembrane α-helical conformation [32]. This observation aligns with the notion that, in solution, MccE492 may adopt an unfavorable helical conformation in comparison to the predominant random coil conformation when analyzed via CD in aqueous solution [13]. Additionally, the high β-sheet content associated with MccE492 amyloid formation was also evidenced by FTIR, where a relatively high alpha-helix content was associated with the non-amyloid forms [17].

Notably, a similar behavior has been described for another amyloidogenic protein when interacting with membranes, alpha-synuclein [33]. Experimental evidence has shown a transition between α-helices and β-sheets in the amyloid formation mechanism of this kind of protein [34]. Furthermore, this concurs with the identification of pentameric precursors of MccE492 bearing the functional structure within the membrane [14]. Adding to this similarity between the two proteins is the structural insight obtained through the Cryo-EM of alpha-synuclein [35], revealing that the beta-sheets in the cross-beta structure are of the parallel type. This convergence of features between both proteins makes comparing their mechanisms of amyloid formation and inhibition highly intriguing.

EGCG is a type of catechin, a class of flavonoids that is commonly found in certain plants, particularly in green tea. Besides the impairment of amyloid formation, this small molecule has been associated with several other health benefits, including antioxidant properties and the ability to neutralize free radicals and protect cells from oxidative damage, potentially reducing the risk of chronic diseases [36]. Also, it has demonstrated potential in cancer prevention and treatment. Preclinical studies suggest that EGCG’s antioxidant properties may protect cells from DNA damage and oxidative stress, factors linked to cancer initiation. Additionally, its anti-inflammatory effects may hinder cancer cell growth and survival. EGCG’s ability to interfere with angiogenesis could inhibit tumor growth and metastasis. Furthermore, it has shown promise in promoting apoptosis in cancer cells and enhancing the effectiveness of chemotherapy and radiation therapy [37,38,39]. Moreover, EGCG was related to increased brain health due to its antioxidant, anti-inflammatory, and anti-aging abilities, as well as its neuroprotective effects [40]. Together, these data place EGCG as a molecule of particular interest, given its proposed versatile and beneficial health effects.

Possible mechanisms have been proposed regarding the effect of EGCG on amyloid formation. It was shown that, when free amines or thiols are close to the EGCG hydrophobic binding sites, the EGCG-based quinones can covalently modify the amyloidogenic proteins through Schiff base formation. The covalent modification of forming amyloid fibers by EGCG can cross-link them, preventing the fragmentation or dissociation required for amyloid propagation and generating toxic oligomeric species of some pathological amyloids [41]. Here, we showed that EGCG also interferes with the early stages of amyloid formation, causing MccE492 to remain soluble and preventing its transition to an amyloid-prone conformation. To further understand the EGCG inhibition of MccE492 amyloid formation, we evaluated if, in the presence of EGCG, MccE492 would maintain its antibacterial activity. We found that, independently of the presence of EGCG, MccE492 lost its activity upon incubation in aggregation conditions (Appendix A). Together, these results suggest that, although EGCG impairs the formation of mature amyloid fibers and SDS-resistant oligomers, it would not affect possible initial conformational changes in the monomers favored by the aggregation conditions, which led to antibacterial activity loss.

EGCG has several reported effects on bacterial amyloids and their related processes. As mentioned, it can inhibit biofilm formation mediated by FapC and curli [42]. Also, it can impair quorum-sensing (QS) signaling by increasing the binding of pyocyanin due to FapC fibril remodeling in *P. aeruginosa*, raising its susceptibility to antibiotics such as tobramycin [23]. Furthermore, EGCG activates the expression of the small non-coding RNA molecule RybB that down-regulates the production of the two main components of *E. coli* biofilm: curli and pEtN-cellulose [24,43]. These examples support the possible use of EGCG as an amyloid-targeting anti-virulence molecule.

As mentioned above, the role of MccE492 in *K. pneumoniae* virulence is not completely clear. In particular, whether MccE492 amyloid formation has implications for the virulence of the producing strains (for instance, forming part of *K. pneumoniae* biofilms) remains unknown. Therefore, it would be interesting to evaluate whether EGCG affects the virulence of MccE492-producing *K. pneumoniae* strains and if this effect is linked to MccE492 amyloid formation.

## 4. Materials and Methods

### 4.1. MccE492 Purification

MccE492 purification was carried out from culture supernatants of *E. coli* BL21 DE3 cells carrying the pMccE492 plasmid, which carries the whole MccE492 gene cluster, as described previously [15]. Briefly, 2 L of M9 medium supplemented with 0.2% citrate and 0.1% glucose were inoculated with a 1:2000 dilution of a fresh overnight culture and grown at 37 °C with shaking at 160 rpm for 18–20 h. The supernatant was collected by centrifugation and filtered through a Stericup 0.22 μm polyethersulfone membrane (Merck, Darmstadt, Germany). The cell-free medium was incubated with 5 g of previously ACN-activated Bondapak C18 resin (Waters, Milford, MA, USA) at 4 °C for 2 h with gentle agitation. Then, the resin was filtered by negative pressure through a Buchner funnel, washed with 100 mL of 40% methanol then with 100 mL of 25% ACN, and finally eluted with a 30–100% ACN stepwise gradient of 50 mL each. MccE492-enriched fractions, as shown in Appendix A, were dialyzed twice for 2 h against 40 nanopure water and then lyophilized and stored at −20 °C.

### 4.2. MccE492 Reconstitution and Preparation for the Aggregation Assays

Lyophilized protein was resuspended in ice-cold 5 mM Tris-HCl pH 8.5 and centrifuged for 30 min at 160,000× *g* at 4 °C. The protein recovered in the supernatant was quantified with the Quick Start Bradford 1X Dye Reagent (Bio Rad, Hercules, CA, USA) following the manufacturer’s instructions. Protein concentration was adjusted to 200 µg/mL in aggregation buffer (100 mM PIPES-NaOH, 0.5 M NaCl pH 6.5) or 400 µg/mL in phosphate buffer (100 mM sodium phosphate pH 6.5).

### 4.3. Aggregation Assays Followed by Thioflavin-T Fluorescence

Aggregation assays were performed as previously described [44]. Briefly, MccE492 samples in PIPES or phosphate buffer were disposed in black 96-well microplates with flat and UV-transparent bottom (4titude 4ti-0263, Azenta, Chelmsford, MA, USA), with or without EGCG (Sigma-Aldrich, Saint Louis, MO, USA), and with 20 µM Thioflavin T (Sigma-Aldrich, Saint Louis, MO, USA), in a final volume of 100 µL per well. The plates were incubated with constant shaking at 37 °C, and ThT fluorescence was measured (in the bottom of the wells) every 15 s using the microplate fluorometer TECAN infinite 200 pro (excitation: 450 nm; emission: 482 nm). Baseline reactions were set up with buffer, with ThT, and with and without EGCG. This value was subtracted from the corresponding samples. The plates were sealed using Microseal B adhesive films (Bio-Rad) to prevent evaporation. Three independent experiments were conducted in each buffer.

### 4.4. Determination of Soluble MccE492 during the Aggregation Assay

After preparing the samples (See Section 4.2), 1 mM EGCG was added when indicated. Samples were incubated with constant shaking (800 rpm) at 37 °C and protected from light during the assay. At different time points, aliquots of the samples were collected and centrifuged for 30 min at 160,000× *g* at 4 °C. The supernatant was recovered, and the remaining soluble protein was visualized by immunoblot.

### 4.5. SDS-PAGE and Immunoblot for MccE492 Detection

Three-phase tricine-sodium dodecyl sulfate-polyacrylamide gel electrophoresis (SDS-PAGE) optimized for peptides was performed as described previously [45]. Nitrocellulose membranes (Millipore, Burlington, MA, USA) were used for immunoblotting transfer (90 min, 400 mA) using chilled 25 mM Tris-HCl, 190 mM glycine, 20% methanol as the transfer buffer. MccE492 was detected using a monoclonal antibody prepared against a synthetic version of the whole MccE492 peptide in mice (1:5000 dilution; GenScript, Piscataway, NJ, USA) and with an anti-mouse IgG alkaline phosphatase-linked secondary antibody (1:5000 dilution; Cell Signaling Technology, Danvers, MA, USA). The alkaline phosphatase colorimetric reaction was carried out using the ready-to-use BCIP^®^/NBT liquid substrate system (Sigma). The membrane was incubated directly with the substrate until an optimal signal was observed.

### 4.6. Negative Staining Electron Microscopy

Aliquots of the samples from the MccE492 aggregation assays (Section 4.4) were mounted onto formvar/carbon-coated copper grids (300-square-mesh; Electron Microscopy Sciences) and negatively stained with 2% uranyl acetate. For staining, each grid was deposited over a 10 µL drop of the aggregation assay mixture, incubated for 30 s, and then removed, drying the excess liquid by contacting the edge of the grid with clean filter paper. Then, a similar procedure was followed, but now using a drop of uranyl acetate solution and incubating for 1 min. Micrographs were taken in a Talos F200C G2 transmission electron microscope operated at 200 kV with a 36,000× magnification (for picture capturing) using the Velox Imaging software (https://veloximaging.com, accessed on 19 October 2023). Several grid fields at different magnifications and regions were observed for TEM analysis to ensure that it correctly represented the samples. Under these conditions, if any amyloid aggregate occurs on the sample, it will be observed. As both kinds of grids (with or without EGCG) have been prepared and observed under equivalent conditions, the absence of fibrillar structures on the sample incubated with the inhibitor observed by TEM reflects that the formation of these structures was prevented. The TEM observations were performed for three independent aggregation experiments (with or without EGCG).

### 4.7. Synchrotron Radiation Circular Dichroism (SRCD)

SRCD measurements were performed on the DISCO beamline at SOLEIL Synchrotron (Saint Aubin, France) [31]. For SRCD experiments, lyophilized MccE492 prepared as described in Section 4.1 was reconstituted in 100 mM phosphate buffer to reach a concentration of 10 mg/mL. A circular CaF_2_ cell of 19 µm path length was used to load the samples (~4 µL), performing spectral acquisitions of 1 nm steps at 1.2 s integration time between 261 and 170 nm, in triplicate. Intensities obtained were calibrated using a (+)-camphor-10-sulfonic acid (CSA) standard solution. Increasing 3 °C steps from 15 °C to 96 °C were carried out for thermal stability measurements. Data averaging, baseline subtraction, smoothing, and scaling were carried out with CdtoolX [46]. Intensity (mdeg) to delta epsilon (M^−1^ cm^−1^) unit conversion was made considering protein concentration calculated from the absorbance at 205 nm and the MccE492 amino acid sequence, as described previously [47]. Protein secondary structure was determined with BeStSel software (https://bestsel.elte.hu/index.php, accessed on 19 October 2023) [27]. Normalized root-mean-square deviation (NRMSD) indicated the most accurate fit for each spectrum; values of <0.15 were considered significant.

## Figures and Tables

**Figure 1 molecules-28-07262-f001:**
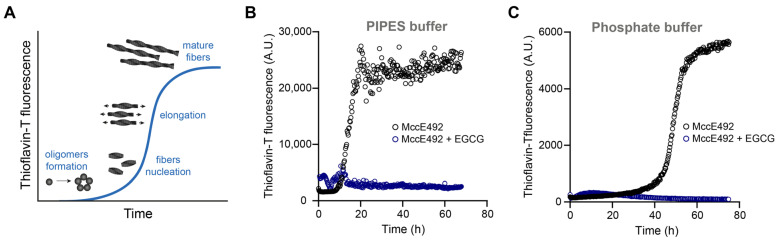
Effect of EGCG on MccE492 amyloid formation monitored by ThT fluorescence. (**A**) Schematic representation of the typical amyloid aggregation kinetics followed by ThT. (**B**) MccE492 (200 µg/mL) aggregation kinetics in PIPES aggregation buffer, with or without 1 mM EGCG. (**C**) MccE492 (400 µg/mL) aggregation kinetics in phosphate buffer, with or without 1 mM EGCG. Amyloid formation was monitored by measuring ThT fluorescence (excitation: 450 nm; emission: 482 nm) every 15 s for 70 h at 37 °C. A.U. = arbitrary units. The curves are representative of three independent experiments with each buffer.

**Figure 2 molecules-28-07262-f002:**
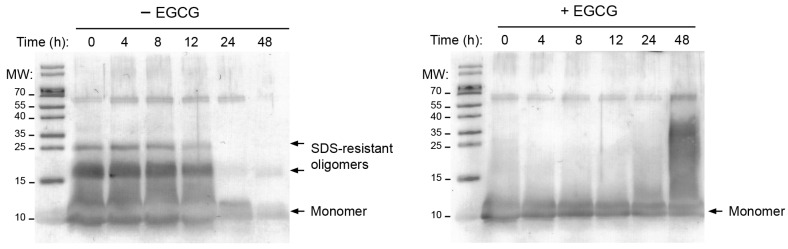
EGCG promotes a soluble state of MccE492 and impairs the formation of higher-order aggregates. Purified MccE492 (200 µg/mL) was incubated in the absence or presence of 1 mM EGCG in PIPES aggregation buffer at 37 °C. The soluble protein was visualized by immunoblot after centrifuging the samples at 16,000× *g* for 30 min and recovering the supernatant. A monoclonal anti-MccE492 antibody was used. MccE492 monomers (~10 kDa) and oligomers (>10 kDa) are pointed out. MW: molecular weight standard (kDa). Although MccE492 has a molecular weight close to 8 kDa, it has anomalous electrophoretic migration, with an apparent MW of 10 kDa.

**Figure 3 molecules-28-07262-f003:**
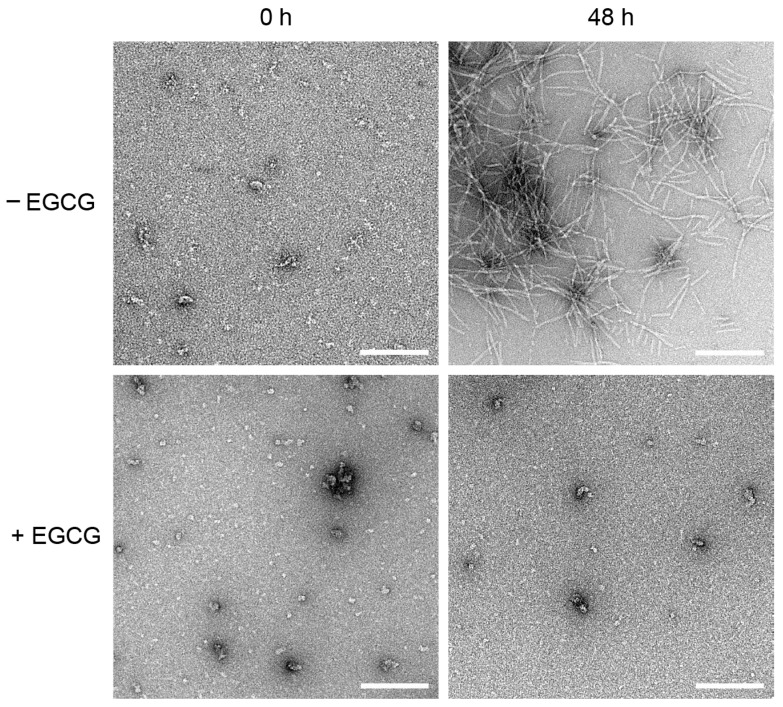
Visualization of MccE492 amyloid fibers morphology in the absence and presence of EGCG. Purified MccE492 (200 µg/mL) was incubated with or without 1 mM EGCG in PIPES aggregation buffer at 37 °C. At the indicated times, samples were collected and visualized by negative-stain electron microscopy. Scale bar: 200 nm. Images are representative of three independent experiments. Additional images are shown in Appendix A.

**Figure 4 molecules-28-07262-f004:**
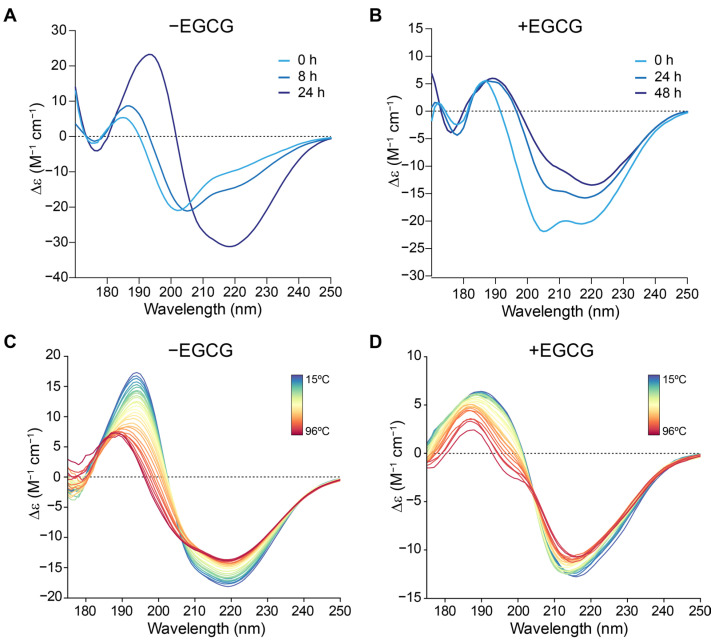
Effect of EGCG on MccE492 amyloid secondary structure signature and thermal stability, as revealed by SRCD. SRCD spectra of MccE492 (2 mg/mL) in phosphate buffer incubated at 37 °C in the absence (**A**) or presence (**B**) of 1 mM EGCG. (**C**,**D**) SRCD thermal scan of preformed amyloid fibers without (**C**) or with (**D**) EGCG 1 mM.

## Data Availability

The data that support the findings of this study are available on request from the corresponding authors.

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
