# Peer review of "The Green Tea Polyphenol Epigallocatechin-Gallate (EGCG) Interferes with Microcin E492 Amyloid Formation"

_molecules, 2023, doi:10.3390/molecules28217262_

Round 1

Reviewer 1 Report

I have had the privilege of reviewing the manuscript titled "The green tea polyphenol epigallocatechin-gallate (EGCG) interferes with microcin E492 amyloid formation" by Aguilera et al. I appreciate the authors' efforts in investigating the inhibitory effects of polyphenol EGCG on the aggregation of Microcin E492. Overall, the study presents valuable insights into amyloid aggregation inhibition. However, I recommend that the authors address several major and minor revisions before considering the manuscript for publication.

Major Revisions:

  1. Quenching Effect on ThT Fluorescence: Some compounds known to inhibit amyloid aggregation may also interfere with ThT fluorescence, leading to misleading results. The authors should address this concern by measuring light scattering as an orthogonal method to detect aggregate formation. This additional data will enhance the reliability of their findings.
  2. Statistical Data and Replicates: The manuscript lacks statistical data, such as error bars, and does not specify the number of replicates in kinetic plots. Including this information is essential for the robustness of the study's conclusions.
  3. Titration and Time-Dependent Analysis: To gain deeper insights into EGCG's inhibitory capacity, I recommend further analysis through titration at increasing compound: protein ratios. Additionally, studying the addition of EGCG at different time points during the aggregation kinetics will provide a more comprehensive understanding of its inhibitory mechanisms.
  4. TEM Confirmation: While the authors state that no fibers were observed in the presence of EGCG, TEM is not suitable for quantification. I suggest performing static light scattering measurements to confirm these observations more accurately.
  5. Secondary Structure Analysis: The manuscript reports changes in secondary structure using SRCD but lacks information on temperature effects on Mcc degradation, and EGCG stability. To address this, the authors should conduct experiments at various temperatures and consider using FTIR spectroscopy to quantify beta-sheet secondary structures.
  6. Alpha-Helix Conformation: The manuscript does not explain why MCC protein exhibits an alpha-helix conformation at zero time in the presence of EGCG, which later transitions to a different structure. Please clarify this point.

Minor Revisions:

  1. Figure Improvements: In Figure 2, it would be beneficial to display the upper part of the gel/blot to detect large aggregates that cannot enter the gel. In Figure 4, please increase the size of axis labels and correct the legend to accurately reference figures.
  2. Glycosylation and Antibacterial Activity: Authors claim that glycosylation is required for antibacterial activity, but the recombinant protein is produced in E. coli. This inconsistency needs to be addressed to ensure clarity.
  3. Materials and Methods section:

To enhance the reproducibility of the aggregation experiments, the following information should be added to the Materials and Methods section:

  • Provide an SDS-PAGE gel image to evaluate the purity of the protein source.
  • Explain why pre-aggregated forms are not removed by filtering before aggregation assays.
  • Specify the final volume of the aggregation reactions.
  • Describe the method used to prevent sample evaporation.
  • Report the type of microplate used in aggregation assays, including the supplier and reference code.
  • Indicate the volume of Mcc added to ThT.
  • Detail the protocol for negative staining for TEM visualization.

Author Response

We thank the reviewers for their criticism and dedication to reviewing our article. Based on their feedback, we prepared an improved manuscript version. Also, we are attaching a point-by-point response to the comments.

Reviewer 2 Report

In this paper The Green Tea Polyphenol Epigallocatechin-Gallate (EGCG) Interferes with Microcin E492 Amyloid Formation, the authors examined the ability of the green tea flavonoid epigallocatechin gallate (EGCG) to interfere with microcin E492 amyloid formation. Aggregation kinetics followed by thioflavin T binding were used to monitor amyloid formation in the presence or absence of EGCG. Additionally, synchrotron radiation circular dichroism (SRCD) and transmission electron microscopy (TEM) were used to study the secondary structure, thermal stability, and morphology of the microcin E492 fibers. The article is in line with the readers interest of molecules. However, there are still some obvious shortcomings presented as follows.

Comments:

Q1. The author has conducted extensive literature research, and the introduction section is highly detailed. However, the core innovative aspects and research significance of this paper remain unclear.

Q2. Figure 1 showed the Effects of EGCG on MccE492 amyloid formation monitored by ThT fluorescence. Are there any related fluorescence images?

Q3. In Figure 2, The soluble protein was visualized by immunoblot after centrifuging the samples at 16000 x g for 30 min and recovering the supernatant. Are these bands in this picture stained with some antibodies?

Q4. SDS-PAGE only showed the Molecular weight distribution of proteins. The determination of the solubility state of different proteins is a subject of inquiry.

Q5. Whenever feasible, it is advisable to complement the potential interaction mechanisms of the EGCG on MccE492 amyloid.

Q6. The paper lacks sufficient effective data, leading to an inadequate workload. It is recommended to supplement additional experiments or data in order to enhance the quality of the paper.

Author Response

(The authors gave the same response as above.)

Round 2

Reviewer 2 Report

No additional comments.